# REFLECTIVE REINFORCEMENT TOOL LEARNING

## ABSTRACT

Tool learning enables large language models (LLMs) to interact with real-world environments. While prior work mainly relies on supervised fine-tuning (SFT), recent reinforcement learning (RL) methods have shown promise in improving the tool-use capabilities of LLMs by leveraging richer reward signals. However, during RL rollouts, failures often stem from environmental perturbations such as network issues or tool instability rather than policy errors. These failed trajectories are typically discarded, resulting in low data efficiency and high costs, especially when using paid tools. To solve the issue, we find that many failures can be recovered through simple retries, reasoning, or reflection. Yet these augmented new policies for self-correction introduce distribution shifts that hinder the reuse of recovered data for origin policy learning. In this paper, we propose **Tool-Reflective Reinforcement Learning** (**Tool-ReRL**), an off-policy RL framework that equips LLMs with a reflection mechanism to temporarily adjust the rollout policy, thus analyzing failures, attempting self-correction, and exploring diverse solution paths. To bridge the distribution gap between modified and original policy, we introduce an importance sampling estimator, enabling rewards from reflection-enhanced trajectories to effectively guide the optimization of the original policy. Our extensive experiments on four tool-learning benchmarks demonstrate that, given the same training data, Tool-ReRL significantly improves data efficiency and achieves average performance gains of up to 7.60% and 6.11% over standard RL algorithms based on Qwen2.5-7B and LLaMA3.1-8B, respectively.

## 1 INTRODUCTION

Tool learning, aiming to enable LLMs to master various external tools, makes LLM-based agents perceive and interact with real-world environments through tools (Baker et al., 2020; Nakano et al., 2022; Qin et al., 2023). Previous methods primarily focus on delicately curating high-quality expert data (e.g., tool-use demonstrations) for supervised fine-tuning (SFT) (Schick et al., 2023; Qin et al., 2024; Qu et al., 2025), where LLMs tend to imitate the demonstrations instead of exploration. This paradigm limits their generalization in open-ended and complex real-world tool-using scenarios. Recent methods adopt reinforcement learning (RL) to mitigate the limitations inherent in SFT by enabling LLMs to interact with the environment through trial-and-error learning (OpenAI et al., 2024; DeepSeek-AI et al., 2025), thereby allowing them to refine their policies based on environmental feedback and learn from more flexible reward signals (Chu et al., 2025).

Existing RL methods for tool learning mainly focus on designing reward mechanisms, such as format compliance (Qian et al., 2025; Singh et al., 2025), call correctness (Feng et al., 2025; Li et al., 2025d), and hierarchical multi-step execution (Dong et al., 2025). These mechanisms implicitly assume that environments are relatively stable and predictable. However, in real-world settings shown in Fig 1, environmental perturbations, such as network instabilities and IP restrictions triggered by exceeding access limits, inevitably generate substantial numbers of failed trajectories. Under current reward mechanisms, these failure trajectories are either treated as uninformative negative examples or, more problematically, may erroneously penalize correct model behaviors (Arnal et al., 2025; Singh et al., 2025). This leads to training inefficiency due to the prevalence of negative samples.

Given the prevalence of failed trajectories and their limited utility in current RL frameworks, a natural question arises: should we simply discard all failed trajectories, or can we differentiate among them to extract valuable training signals? To address this fundamental challenge, we first investigate the value and characteristics of failed trajectories in tool learning scenarios. Our analysis reveals that

Figure 1: An illustration of current reinforcement learning (RL)-based tool learning methods shows that the rollout processes are frequently interrupted by environmental perturbations, such as network fluctuation, API limits, Invalid Files and IP change. Since these errors are external to the policy, they are commonly excluded from training. This practice, while preventing unfair penalization of the policy, inadvertently leads to the systematic discarding of trajectories, thereby limiting data efficiency and inflating training costs.

beyond cases that exceed model capabilities, a substantial portion of failed trajectories represent near-success instances where the policy executes most steps correctly but fails due to minor oversights or environmental interruptions. Specifically, in a dataset (Mialon et al., 2024) of 2,400 trajectories generated by Qwen3-32B (Yang et al., 2025), we identified 834 failures in total, of which 373 (44.7%) were attributable to near-success or environment-related issues. This suggests that nearly half of the failed trajectories are not fundamentally erroneous. For these cases, the model requires only minimal reflection or simple retry mechanisms to transform these previously unusable negative samples into valuable positive training signals. This insight leads to our core design principle: instead of discarding near-success failures, we should actively repair them within the training loop. To achieve this, we need a framework that can learn from corrected trajectories generated by a "reflection" policy, while ensuring the stability of the main learning process.

In this paper, we propose **Tool-Re**flective **R**einforcement **L**earning (**Tool-ReRL**), an off-policy online RL framework designed to effectively utilize these near-success failures. At its core, Tool-ReRL integrates a novel online correction mechanism. Unlike previous methods that perform correction offline, our approach invokes a temporary, reflection-driven policy to repair trajectories within a single RL training loop. A key challenge arises from this design: learning from data generated by this alternate reflection policy can introduce a distribution mismatch, destabilizing the training of the original target policy. To address this, Tool-ReRL employs importance sampling (Precup, 2001; Degris et al., 2012; Schulman et al., 2015), a theoretically-grounded technique that re-weights the corrected trajectories. This allows the target policy to safely and efficiently absorb the valuable signals from repaired failures, ensuring stable and unbiased policy updates.

Specifically, we first propose an online reflection strategy that enables LLMs to perform self-criticism and reflection during RL, thereby repairing failed trajectories by generating reflection-augmented queries that transform previously discarded rollouts into valuable training signals. Second, we construct an importance-sampling-based estimator that re-weights these reflection-driven trajectories, aligning them with the original on-policy distribution. This dual design enables the original policy to benefit from repaired failures while maintaining unbiased updates and stable training dynamics, thereby improving both data efficiency and overall effectiveness.

The extensive experiments on four popular used tool-learning benchmarks demonstrate that, given the same training data, Tool-ReRL significantly improves data efficiency and achieves average performance gains of up to 7.60% and 6.11% over standard RL algorithms on Qwen2.5-7B and LLaMA3.1-8B, respectively.

## 2 RELATED WORK

**Reinforcement Learning for LLMs.** Recently, the o1 and R1 models have garnered substantial attention due to their remarkable task-solving capabilities, achieving strong reasoning performance

that surpasses that of humans on popular mathematical and coding benchmarks (Shao et al., 2024; DeepSeek-AI et al., 2025; OpenAI et al., 2024; Yang et al., 2025). This success attracts extensive research endeavors leveraging RL to enhance various aspects of LLM capabilities, resulting in numerous successful models (Hu et al., 2025; Chen et al., 2025; Cheng et al., 2025; Xiang et al., 2025). Despite these achievements, most approaches in RL training continue to face constraints related to positive sample search success rates and data efficiency (Zelikman et al., 2022; Gulcehre et al., 2023; Hosseini et al., 2024; Kumar et al., 2024). Specifically, during the rollout phase, if the policy lacks sufficient capability to sample positive examples with reasonable success rates, its contribution to RL effectiveness becomes negligible. To address this challenge, research focusing on mathematical and coding domains has employed negative feedback signals to penalize model failures, thereby encouraging models to explore correct solution paths and achieve significant improvements (Shao et al., 2024; Zheng et al., 2025; Hu et al., 2025). While these domains typically feature unique correct answers and singular error sources caused by LLMs themselves, penalizing models solely based on incorrect final outcomes is inappropriate in tool-calling scenarios, as error sources are diverse and may stem from environmental perturbations (such as API key limits, network instabilities, file system changes, etc.) rather than policy deficiencies. Penalizing models based solely on incorrect results often leads to policy collapse (Arnal et al., 2025; Singh et al., 2025). Consequently, negative samples in tool-calling scenarios are frequently discarded, which exacerbates RL data efficiency issues and introduces high computational costs. In this paper, we differentiate between negative samples and extract valuable training signals through an online reflection mechanism, which enhances the effectiveness and efficiency of RL.

**Tool Learning with LLMs.** The tool learning domain has experienced rapid development driven by the remarkable advancement in LLMs' language understanding capabilities (Nakano et al., 2022; Yao et al., 2023; Surís et al., 2023; Gou et al., 2024; Gao et al., 2024). Most previous methods employ SFT to enhance the tool learning abilities of models (Schick et al., 2023; Hao et al., 2023; Qin et al., 2024). They typically construct training data by sampling from the trajectories of stronger models, thereby scaling both data and training effectiveness and yielding numerous well-performing models (Qin et al., 2024; Liu et al., 2025). Recently, the tremendous success of o1 and R1 has sparked a surge of interest in exploring the effectiveness and efficiency of reinforcement learning in the tool learning domain (Jin et al., 2025; Feng et al., 2025; Singh et al., 2025). Existing reinforcement learning approaches for tool learning can be categorized into two paradigms: the first focuses on maximizing the value of positive samples through sophisticated reward design (Qian et al., 2025; Singh et al., 2025; Li et al., 2025d), external models such as LRMs (Li et al., 2025a; Wu et al., 2025a; Li et al., 2025b), or enhanced reasoning processes (Jin et al., 2025; Li et al., 2025c; Dong et al., 2025) to extract maximum learning from successful trajectories. The second category draws inspiration from preference learning methodologies, such as DPO (Li et al., 2025c; Dong et al., 2025), which employs contrastive learning that leverages both successful and failed trajectories to guide the learning process. However, these approaches incur substantial costs and data efficiency challenges due to the unavoidable environmental perturbation. In this paper, we propose Tool-ReRL to repair failed trajectories through online reflection within a single RL training loop, which enhances both data efficiency and effectiveness of RL for tool learning.

## 3 TOOL-RERL

In this section, we introduce our proposed novel Tool-Reflective Reinforcement Learning framework (**Tool-ReRL**), aiming at improving both the efficiency and effectiveness of RL training for LLMs in the tool learning domain. We begin by formalizing the tool learning task as a reinforcement learning problem (Section 3.1), establishing the foundation for policy-based optimization. To address the inefficiency of standard RL approaches that discard failed interactions during data collection, we introduce a New Reflection-Recovery Strategy (Section 3.2) that transforms such waste trajectories into valuable training signals. However, this introduces off-policy data and induces a distributional shift. Such a shift, if left unaddressed, can severely compromise both the stability and the convergence of policy optimization toward the intended learning objective. To address the distributional shift introduced by reflection-augmented trajectories, we introduce our Importance-Weighted Correction mechanism (Section 3.3), which re-weights each trajectory based on its likelihood under the current policy. This ensures alignment with the on-policy distribution while preserving the informational value of reflection-based feedback.

Figure 2: An illustration of the Tool-ReRL framework. Top: Standard PPO-based training, where the policy model generates a trajectory ($o$) from an original query ($q$). Bottom: Our proposed Tool-ReRL framework. For each failed trajectory during data collection, a reflection model generates an explanatory reflection concatenated after $q$ and its historical output to form an augmented query $q^*$. During training, the policy generates two trajectories, one conditioned on $q$ and the other on $q^*$. The key innovation lies in the use of an Importance weight, which reweights the advantage based on the likelihood ratio.

## 3.1 TASK FORMULATION

Following prior work, we formulate tool learning as a reinforcement learning problem where the language model, governed by a policy $\pi_\theta$, interacts with an external environment over multiple steps. At each step $t$, the state $s_t$ includes the user query and interaction history; the model then samples an action $a_t$ such as an intermediate reasoning step or a structured tool call. The environment returns an observation $o_{t+1}$ and a reward $R_t$, producing a trajectory $\tau = (s_0, a_0, \ldots, s_T, a_T)$. For later analysis, we also use $\tau_i = (t_i, a_i)$ to denote a *trajectory fragment* consisting of a reasoning step $t_i$ and a tool call $a_i$ at step $i$, conditioned on the current query $q_i$. The learning objective is to maximize the expected return:

$$\max_\theta \ \mathbb{E}_{\tau \sim \pi_\theta} \left[ \sum_{t=0}^{T} \gamma^t R_t \right], \tag{1}$$

with discount factor $\gamma \in [0, 1]$. This formulation serves as the foundation for the Tool-ReRL framework, which aims to optimize tool-use behavior in language models through interaction-driven learning.

## 3.2 THE REFLECTION-RECOVERY STRATEGY

In tool-use scenarios, many trajectory failures stem not from policy errors but from external factors such as API limits, network disruptions, or tool instability. These failures, which can often be identified through abnormal tool responses or known error codes, are typically discarded in conventional RL pipelines—leading to significant data inefficiency.

To address this, we propose a *Reflection-Recovery Strategy* that transforms such failed interactions into training signals. Specifically, for each failed trajectory $\tau$, we retain the interaction context and

invoke a separate reflection module. This module analyzes the failure and generates a reflection $rfl$ that contains a diagnosis and a suggested remedy.

We then construct an augmented query $q^* = [q, t, a, rfl]$ by concatenating the original user query $q$, the model's thought $t$, its failed action $a$, and the reflection $rfl$. This enriched input is used to guide the policy model $\pi_\theta$ in generating a revised trajectory $\tau^*$ that addresses the original error.

Although these corrected trajectories provide valuable supervision, they are generated under altered input conditions not present during standard inference. As such, they are inherently *off-policy* with respect to the original query distribution. In the next section, we discuss how to incorporate these off-policy samples via importance-weighted correction.

### 3.3 IMPORTANCE-WEIGHTED CORRECTION

As discussed in the previous section, the construction of a corrected whole multi-step trajectory $\tau^*$ is conditioned on an augmented query $q^*$, which includes externally generated reflection text. As a result, these trajectories are *off-policy* with respect to the original policy $\pi_\theta$ we aim to optimize. Naively training on $\tau^*$ would lead to a severe distributional shift. To correct this mismatch, we employ importance weighting to reweight each training signal based on the likelihood ratio between the target policy and the behavior policy that generated the data. We first formalize the underlying distributions as follows:

**Definition 1.** *We define two trajectory sampling processes: i. $\pi_\theta(\tau_i) = \pi_\theta(t_i, a_i \mid q_i)$ denotes a fragment trajectory sampled from policy $\pi_\theta$ conditioned on the each query $q_i$, where $t_i$ is the intermediate reasoning and $a_i$ is the tool call. ii. $\pi_\theta(\tau_i^*) = \pi_\theta(t_i^*, a_i^* \mid q_i^*)$ denotes a trajectory sampled under the same policy conditioned on an augmented query $q^* = [q, t, a, rfl]$, which includes historical reasoning steps and a reflection generated by another LLM. In practice, $\tau^*$ is obtained via iterative sampling with early stopping: at each round, the model generates a candidate under $q^*$, and the process stops once a valid trajectory is found or a maximum number of attempts is reached.*

Note that while we introduce $\tau_i$ to denote trajectory fragments for the convenience of analysis, the final policy optimization objective is still defined over full trajectories $\tau$.

**Assumption 1.** *The existence of a behavior policy $\pi_{\theta_{rfl}}$—a parameterization of the same model that can generate improved trajectories $\tau^*$ when conditioned on the original query $q_i$ for all queries in a multi-step task :*
$$\pi_{\theta_{rfl}}(\tau_i^*) = \pi_{\theta_{rfl}}(t_i^*, a_i^* \mid q_i).$$
*Since the actual data is collected by sampling $\tau^*$ under an augmented prompt $q^*$ using the current policy $\pi_\theta$, we make the following approximation:*
$$\pi_{\theta_{rfl}}(t_i^*, a_i^* \mid q_i) \approx \pi_\theta(t_i^*, a_i^* \mid q_i^*).$$
*This inverse prompt equivalence assumption enables us to utilize the tractable quantity $\pi_\theta(\tau^* \mid q^*)$ to approximate the behavioral policy when computing importance weights, without requiring explicit access to $\pi_{\theta_{rfl}}$.*

Consequently, the optimization objective of TOOL-RERL is:

$$\max_\theta \quad \mathbb{E}_{\tau^* \sim \pi_{\theta_{rfl}}} \left[ \min \left( \frac{\pi_\theta(\tau^*)}{\pi_{\theta_{rfl}}(\tau^*)} \hat{A}(\tau^*), \ \text{clip}\left( \frac{\pi_\theta(\tau^*)}{\pi_{\theta_{rfl}}(\tau^*)}, \ 1 - \epsilon, \ 1 + \epsilon \right) \hat{A}(\tau^*) \right) \right] \quad (2)$$

Where $\hat{A}(\tau^*)$ denotes the estimated advantage of $\tau^*$, computed following standard PPO practice. As demonstrated in Section 4.2, this correction is both *necessary* and *sufficient* for transforming failures into reliable learning signals—leading to consistent performance gains across all benchmarks. More details are presented in the Appendix A.2.

## 4 EXPERIMENT

### 4.1 EXPERIMENTAL SETTINGS

**Experimental Configuration.** We select Qwen-2.5-7B-Instruct and Llama-3.1-8B-Instruct as our base models. Both models are loaded in FP16 precision on eight A100-80GB GPUs. For all

PPO experiments, we configure the training with a total batch size of 1024 and a mini-batch size of 256. Since tool invocation does not require long-form reasoning, each generated sequence is limited to 512 tokens—sufficient to accommodate a brief Chain-of-Thought followed by the function call—thereby avoiding the excessive latency and computational cost commonly associated with Large-Reasoning-Model style long rollouts.

**Baselines** To assess the effectiveness of our method, we compare it against seven competitive baselines, each representing a distinct training paradigm. All models are fine-tuned or trained on the same ToolACE dataset to ensure fairness. *Base* refers to the pre-trained model without any additional training. *SFT* applies supervised fine-tuning using LoRA on the open-sourced ToolACE dataset, following the hyperparameters specified in the original paper. *DPO* is trained on failure-success pairs, where each query includes an initial invalid call followed by a valid call obtained after a single reflection retry. *PPO* denotes standard Proximal Policy Optimization trained on ToolACE data. In this setting, the model is expected to directly generate the final tool call, aligning with the ToolACE format without requiring intermediate reasoning. *+CoT* augments PPO by requiring rollouts to emit an explicit Chain-of-Thought before the final function call. *+Ref* uses reflection-augmented successful trajectories but naïvely treats them as on-policy, without correcting for distributional shift. *+CoT+IS* allows the model to generate its own internal *thought* sequence but does not perform reflection, while applying importance weights to reduce distributional discrepancy. Finally, Tool-ReRL constitutes our complete framework: failure cases are transformed into reflection-augmented trajectories, which are then incorporated into policy learning via importance-weighted correction, enabling PPO to benefit from off-policy data. All baselines from +CoT onward—including +Ref, +CoT+IS, and Tool-ReRL—adopt a consistent output format during training: the model is required to first generate an intermediate reasoning step (thought), followed by a structured tool call asked by the dataset. This ensures comparability across methods. The primary differences among these variants lie in the construction of the input (e.g., use of reflection) and whether importance weighting is applied to mitigate distributional shift. For clarity, the *IS* used in our baselines denotes the extra importance-sampling correction that our framework applies to compensate for distributional shift introduced by query-level modifications; it is distinct from the standard PPO ratio used within the policy update.

## 4.2 MAIN RESULTS

Table 1 summarizes the performance of all baseline and proposed methods across four benchmarks. As expected, *Base* serves as the lower bound, reflecting model performance without any task-specific adaptation. *SFT* yields modest gains for Qwen-2.5, but interestingly leads to a slight performance drop for LLaMA-3.1. Suggesting that supervised fine-tuning may not generalize well across models with differing pretraining distributions. We adopt *PPO* as our primary training method for two key reasons. First, it delivers the most substantial improvements under sparse reward conditions, which are inherent to tool-use scenarios where learning signals are only available upon successful execution. As discussed in Section 3.1, tool learning is naturally framed as a reinforcement learning problem driven by outcome-based rewards. Second, unlike DPO, PPO does not require evaluating both successful and failed responses for each query, thereby halving the real API costs as we dont need both successful and failed trajectories while focusing updates on genuinely erroneous trajectories.

### 4.2.1 THE NECESSITY OF IMPORTANCE-WEIGHTED CORRECTION

To better understand the challenges posed by reflection-augmented trajectories and to motivate the necessity of importance-weighted correction, we conduct a controlled comparison using rollouts with explicit Chain-of-Thought reasoning (*+CoT*) as a baseline for analyzing distributional drift. When these self-generated thoughts are replaced with externally provided reflections (*+Ref*), the resulting inputs become substantially more informative. However, on Qwen, the average performance drops from 54.94% with +CoT to 51.54% with +Ref, a decline of 3.4%. LLaMA-3.1 shows no notable difference: 46.22% with +CoT versus 46.21% with +Ref. These results suggest that, despite their semantic richness, reflection-augmented inputs may diverge significantly from the original training distribution, thereby impairing reinforcement learning performance and limiting generalization. In the absence of correction mechanisms such as importance weighting, the naive incorporation of such inputs can be ineffective or even detrimental.

Table 1: Model performance on various tool learning benchmarks. All the models are trained on the open-sourced ToolACE training data.

| | | | Qwen2.5-7B | | | | | | | | Llama3.1-8B | | | | | | | |
| | | | Base | SFT | DPO | PPO | +CoT | +Ref. | +CoT+IS | Tool-ReRL | Base | SFT | DPO | PPO | +CoT | +Ref. | +CoT+IS | Tool-ReRL |
|---|---|---|---|---|---|---|---|---|---|---|---|---|---|---|---|---|---|---|
| RotBench | CLE | Par. | 25.71 | 33.33 | 22.86 | 56.19 | 58.10 | 49.52 | 60.95 | **63.86** | 22.86 | 0.95 | 20.95 | 48.57 | 40.00 | 36.19 | 41.90 | **53.33** |
| | | Cont. | 20.00 | 26.67 | 17.14 | 44.76 | 46.67 | 40.95 | 45.71 | **51.43** | 17.14 | 0.00 | 15.24 | 36.19 | 31.43 | 30.48 | 35.24 | **40.00** |
| | HEA | Par. | 21.90 | 20.00 | 21.43 | 30.00 | 37.62 | 30.00 | 39.05 | **45.24** | 13.33 | 0.95 | 13.33 | 33.33 | 25.71 | 24.29 | 26.19 | **36.19** |
| | | Cont. | 14.29 | 16.19 | 14.29 | 22.38 | 30.00 | 22.86 | 29.52 | **32.86** | 8.10 | 0.48 | 7.62 | 21.90 | 16.67 | 16.19 | 17.14 | **23.81** |
| | MED | Par. | 25.71 | 23.33 | 23.33 | 49.52 | 55.24 | 45.71 | 58.10 | **59.05** | 15.71 | 4.29 | 17.14 | 46.67 | 37.14 | 35.71 | 43.33 | **53.81** |
| | | Cont. | 20.00 | 20.48 | 18.57 | 39.05 | 43.81 | 34.76 | 40.00 | **48.57** | 11.43 | 3.81 | 12.86 | 33.81 | 29.52 | 29.05 | 34.29 | **39.05** |
| | SLI | Par. | 24.76 | 29.52 | 23.33 | 50.00 | 51.90 | 42.38 | 57.62 | **60.95** | 16.67 | 2.38 | 18.10 | 40.95 | 34.76 | 30.95 | 36.19 | **47.62** |
| | | Cont. | 19.52 | 23.81 | 18.57 | 39.52 | 41.43 | 33.33 | 43.33 | **50.00** | 12.86 | 1.90 | 14.29 | 29.05 | 28.10 | 25.71 | 28.57 | **33.81** |
| | UNI | Par. | 20.95 | 20.95 | 24.76 | 38.10 | 45.71 | 34.29 | 45.71 | **53.33** | 14.29 | 6.67 | 13.33 | 36.19 | 29.52 | 29.52 | 34.29 | **44.76** |
| | | Cont. | 16.19 | 24.76 | 20.00 | 26.67 | 37.14 | 25.71 | 36.19 | **42.86** | 9.52 | 5.71 | 9.52 | 21.90 | 22.86 | 21.90 | 27.62 | **32.38** |
| TaskBench | HF | n_f1 | 65.80 | 61.18 | 65.63 | 64.35 | 66.88 | 66.74 | 63.80 | **68.70** | 59.20 | 61.18 | 58.75 | 61.93 | 61.64 | 64.13 | 65.13 | **67.51** |
| | | t_f1 | 60.50 | 56.91 | 60.36 | 59.21 | 61.80 | 61.79 | 59.10 | **63.54** | 48.70 | 49.37 | 48.30 | 53.99 | 53.71 | 56.46 | 58.30 | **61.64** |
| | | v_f1 | 38.20 | 36.93 | 37.94 | 38.60 | 39.52 | 38.97 | 37.33 | **40.78** | 22.20 | 23.69 | 21.84 | 29.12 | 29.41 | 30.41 | 32.69 | **38.76** |
| | | l_f1 | 17.30 | 16.66 | 17.43 | 16.36 | 17.33 | 17.68 | 16.82 | **18.59** | 18.10 | 16.92 | 17.89 | 18.70 | 18.99 | 19.17 | 18.80 | **19.62** |
| | Mm | n_f1 | 79.90 | 77.64 | 79.72 | 80.59 | 81.62 | 81.14 | 81.15 | **82.32** | 69.10 | 77.73 | 68.34 | 72.47 | 72.76 | 75.62 | 77.95 | **77.78** |
| | | t_f1 | 74.50 | 73.28 | 74.38 | 75.14 | 76.39 | 76.11 | 76.57 | **77.18** | 59.70 | 62.95 | 58.99 | 66.96 | 67.46 | 70.72 | 72.94 | **73.18** |
| | | v_f1 | 48.20 | 48.62 | 48.38 | 49.91 | 48.91 | 49.85 | 48.69 | **50.49** | 33.10 | 30.98 | 32.49 | 41.31 | 42.02 | 43.85 | 45.24 | **47.76** |
| | | l_f1 | 31.20 | 30.26 | 31.12 | 31.57 | 32.12 | 31.69 | 32.14 | **32.54** | 29.90 | 32.16 | 29.95 | 32.23 | 32.16 | 32.97 | 33.05 | **33.70** |
| BFCL | Non-Live | Ast | 85.46 | 85.12 | 86.02 | 85.96 | 86.02 | 86.48 | 86.25 | **86.56** | 83.90 | 81.50 | 84.19 | 77.75 | 84.83 | 85.65 | 85.17 | **85.67** |
| | Live | Ast | 74.76 | 77.42 | 75.57 | 76.83 | 78.39 | 77.87 | 78.68 | **79.20** | 72.54 | 71.21 | 72.35 | 71.65 | 72.61 | 71.43 | 71.87 | **72.76** |
| Seals | Total | Avg. | 91.41 | 92.69 | 91.39 | 75.37 | 84.19 | 92.43 | 92.86 | **92.99** | 88.69 | 92.26 | 89.31 | 81.56 | 92.29 | 92.47 | 92.65 | **93.17** |
| | OOD | Avg. | 92.76 | 93.30 | 92.77 | 77.30 | 86.90 | 93.80 | 93.92 | **94.35** | 90.93 | 92.85 | 90.78 | 81.45 | 93.22 | 93.70 | 93.90 | **94.47** |
| Total Avg. | | | 44.09 | 45.35 | 43.86 | 51.24 | 54.94 | 51.54 | 55.61 | **58.84** | 37.18 | 32.72 | 37.07 | 47.10 | 46.22 | 46.21 | 48.79 | **53.21** |

Training directly on reflection trajectories without any distribution correction (*+Ref*) yields an impressive boost Pass@1 rises by 7.45% on Qwen-2.5 and 9.0% on Llama-3.1 relative to the base model, but this advantage rapidly diminishes when compared to other PPO-based baselines. As training progresses, the policy overfits to the *query + reflection* input pattern; once the reflection prefix is removed during evaluation, the model's performance is substantially limited by this input-output mismatch, revealing a failure to generalize effectively. In contrast, our Tool-ReRL variant re-weights each reflection trajectory with importance-weighted correction, gradually aligning them with the on-policy query distribution. This correction preserves the informative value of reflection while mitigating drift, leading to both higher final accuracy and greater stability, making Tool-ReRL the strongest performer across all benchmarks and model backbones.

### 4.2.2 THE VALUE OF EXTERNAL REFLECTION

To decouple the contribution of external reflection content, we construct a variant *CoT + IS* in which the augmented query is formed by appending the model's own self-generated thought, rather than an externally provided reflection. Like *Ref*, this variant retains the importance-weighted correction scheme but excludes any additional feedback or retry signals. This leads to a modest 1% average improvement over vanilla PPO on select datasets, suggesting that self-generated reasoning offers a weak yet non-negligible training signal. However, without external feedback, these thoughts often contain unverifiable assumptions or incorrect parameter usage, limiting their ability to surface deeper failure modes or guide effective recovery. Substantial improvements emerge only when failure-driven reflection is reintroduced in conjunction with importance-weighted correction. This reinforces two key conclusions: (i) importance weighting is necessary—without it, all reflection-based variants

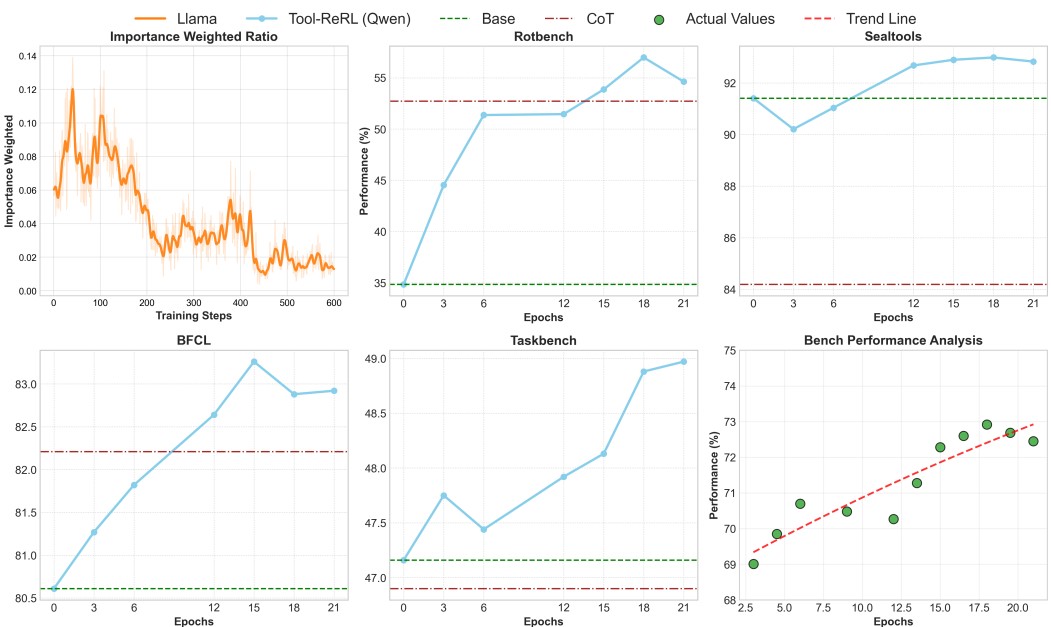

Figure 3: Importance weight correction and benchmark performance of Tool-ReRL. Top left: evolution of importance weights over training steps for LLaMA, demonstrating effective correction of off-policy bias via the importance weighting strategy. Top right and bottom left: Tool-ReRL's performance trends across four tool-use benchmarks, showing consistent improvement and surpassing baselines such as +CoT. Bottom right: aggregated performance over training, with a fitted quadratic trend showing a strong positive correlation between epochs and benchmark scores—highlighting Tool-ReRL's ability to generalise across diverse tool-use tasks.

exhibit significant performance degradation, highlighting that naive incorporation of additional signals without correction only amplifies distributional bias and instability; and (ii) reflection is sufficient—when paired with importance correction, the additional failure-derived signal yields decisive gains. Taken together, these findings demonstrate that importance weighting effectively corrects for distributional mismatch, while external reflection provides the critical feedback required to convert this correction into stable and substantial policy improvements.

### 4.2.3 ANALYSIS OF TRAINING DYNAMICS

Fig. 3 illustrates how importance score weights evolve as off-policy reflection data are integrated into training. For Llama, during steps 0–100, there is a substantial divergence between the on-policy behaviour and the reflection-induced distribution, as indicated by IS weights peaking around 0.12. This reflects a significant mismatch between the original policy and the reflection-conditioned trajectories. As training proceeds and IS-weighted gradients accumulate, the policy rapidly re-aligns: importance weights fall below 0.02 by step 200 and remain consistently low, with a brief adjustment phase between steps 350–420 before stabilising near 0.015. This sharp rise-and-decay pattern suggests that the initial distribution shift, although large, is effectively corrected—preventing overfitting to off-policy inputs.

The subsequent four plots show Tool-ReRL's performance across four diverse benchmarks: RotBench, SealTools, BFCL, and TaskBench. Across all tasks, the blue line representing Tool-ReRL exhibits a steady upward trend over training epochs. Notably, on RotBench and SealTools, Tool-ReRL rapidly outperforms strong baselines such as *+CoT* and continues to improve. The trend is especially steep on TaskBench, highlighting the model's ability to progressively learn from reflection-augmented feedback and refine its tool-use capabilities.

Finally, the bottom-right plot aggregates performance across all benchmarks. The scattered points and the fitted quadratic trend line reveal a strong positive correlation between training progression

Table 2: Comparison between different thought providers.

| Model | Method | RotBench | | | | | TaskBench | | BFCL | | Seals | Total |
|-------|--------|------|------|------|------|------|------|------|------|------|------|------|
| | | CLE | HEA | MED | SLI | UNI | HF | Mm | Non-Live | Live | Total | Avg. |
| **Qwen** | External + IS | 31.43 | 24.05 | 31.43 | 30.96 | 26.67 | 46.35 | 59.06 | 86.87 | 76.09 | 92.67 | 50.56 |
| | External | 25.72 | 22.62 | 29.53 | 28.33 | 26.67 | 44.76 | 57.30 | 86.46 | 75.87 | 92.01 | 48.93 |
| | Internal + IS | 53.33 | 34.29 | 49.05 | 50.48 | 40.95 | 44.26 | 59.64 | 86.25 | 78.68 | 92.86 | 58.98 |
| | Internal | 26.67 | 19.76 | 27.15 | 25.48 | 25.24 | 44.50 | 57.74 | 85.94 | 75.35 | 91.93 | 47.98 |
| **Llama** | External + IS | 29.53 | 19.29 | 29.05 | 29.53 | 24.29 | 41.67 | 52.51 | 84.56 | 73.28 | 90.28 | 47.40 |
| | External | 20.00 | 10.24 | 16.91 | 15.95 | 10.00 | 39.06 | 48.92 | 84.42 | 72.91 | 89.93 | 40.83 |
| | Internal + IS | 38.57 | 21.67 | 38.81 | 32.38 | 30.96 | 43.73 | 57.29 | 85.17 | 71.87 | 92.65 | 51.31 |
| | Internal | 15.72 | 9.52 | 14.05 | 13.10 | 10.00 | 31.26 | 42.66 | 84.67 | 70.24 | 87.00 | 37.82 |

and overall effectiveness, further confirming the generalizability and efficiency of Tool-ReRL across heterogeneous tool-use domains.

## 4.3 ABLATION STUDY

In the previous sections, we analyzed the effectiveness of reflection and importance-weighted correction, as well as training dynamics, respectively. Here we present a unified ablation in Table 2 disentangling the contributions of reasoning provenance and importance weighting.

For the reasoning source, we evaluate the performance of two options: (1) internally generated thoughts produced by the policy model itself, corresponding to the `+CoT + IS` setting; and (2) externally provided reasoning generated by a strong open-source LLM, DeepSeek-R1 (DeepSeek-AI et al., 2025). The external CoT traces provide step-by-step reasoning about how to answer the current query. This contrasts with the richer "reflection" signal used in our full Tool-ReRL framework, which explicitly analyses and corrects failures made in prior trajectories—offering diagnostic feedback rather than direct problem-solving. For each reasoning source, we assess performance both with and without the proposed importance-weighted correction mechanism. The results show that importance weighting significantly improves performance in all cases. When applied, the model using self-generated thoughts (`Internal + IS`) achieves slightly better aggregate performance than the one using external reasoning (`External + IS`). This suggests that, once distributional mismatch is corrected, internal reasoning may align more closely with the model's latent policy, offering a natural integration into the decision-making process.

By contrast, removing the correction mechanism causes a consistent and substantial drop in accuracy, regardless of whether the reasoning is internal or external. These findings highlight that the key factor driving performance is not the origin of the reasoning itself, but rather whether distributional correction is applied. This reinforces our central claim: auxiliary reasoning can be beneficial, but only when properly integrated via mechanisms such as Importance-Weighted Correction.

## 5 CONCLUSION

In this paper, we propose Tool-ReRL to address the fundamental challenges of low data efficiency and high sunk costs inherent in RL-based tool learning methods. Tool-ReRL leverages a reflection mechanism to temporarily modify the policy of LLM, enabling successful sampling of positive examples. The acquired rewards are subsequently transferred to the original policy, thereby maximizing the utility of failure data and enhancing policy sampling efficiency. In future work, we aim to explore transferring reasoning from successful trajectories generated by stronger external models. With importance-weighted correction, such cross-model reflection transfer may enable the efficient reuse of external knowledge, thereby improving both sample efficiency and generalization. Ultimately, this

work points toward a broader paradigm where reflection-driven correction becomes a core principle for building data-efficient and generalizable reinforcement learning systems.

ETHICS STATEMENT

All ICLR participants, including authors, are required to adhere to the ICLR Code of Ethics. All authors have read and agreed to abide by the ICLR Code of Ethics. This work involves reinforcement learning experiments conducted entirely on publicly available benchmark environments and synthetic data. No human subjects, private or personally identifiable information, or sensitive attributes are used. The research does not raise foreseeable concerns regarding fairness, discrimination, privacy, security, or potential societal harm.

REPRODUCIBILITY STATEMENT

It is important that the work published in ICLR is reproducible. We have taken several measures to ensure the reproducibility of our work. In the main paper, we provide detailed descriptions of the reinforcement learning setup, including policy architectures, training algorithms (PPO and Tool-ReRL), and hyperparameters in Section 4.1. Additional theoretical derivations of the algorithms are included in the appendix A.2. All datasets and tool-use benchmarks employed in our experiments are publicly available. To further support replication, we will release an anonymized GitHub repository containing our source code, training configurations, and scripts.

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

# A APPENDIX

## A.1 DISCLOSURE OF USE OF LARGE LANGUAGE MODELS (LLMS)

Large Language Models were employed exclusively for proofreading and language refinement of the manuscript. Their use was limited to improving clarity, grammar, and style to ensure the paper meets academic writing standards. It did not play any role in shaping the research questions, designing the methodology, or conducting the analysis. All substantive contributions remain the sole work of the authors.

## A.2 PPO IN TOOL USING

Proximal Policy Optimization(PPO) consists of two parts: an actor-critic architecture and a gradient clip technique. The actor-critic architecture of PPO consists of two models, an actor model $\pi_\theta$ and a critic model $\pi_\sigma$. The actor $\pi_\theta$ receives the query $q$, a token-level sequence, from the user as the state, and then generates a response sequence $o$ (token-level). The critic network model generates a value $v_\sigma(q, o_t)$ to evaluate the produced action policy and iterates according to the Bellman equation 3 at position $t$:

$$\sigma_{new} = \sigma_{old} + \nabla_\sigma \frac{1}{2} [v_\sigma(q, o_t) - (\gamma * r_t + v_\sigma(q, o_{t+1}))]^2 \tag{3}$$

where $\sigma$ denotes parameters in the critic network and $v_\sigma(q, o_t)$ denotes Q value for agent response at position $t$. Then the actor network leverages the general advantage function 4 to train the actor model:

$$\hat{A}(q, o_t) = \sum_{l=0}^{T-t} (\gamma\lambda)^l A_{t+l} = \sum_{l=0}^{T-t} (\gamma\lambda)^l (r(q, o_{t+l}) + v_\sigma(q, o_{t+l+1}) - v_\sigma(q, o_{t+l})) \tag{4}$$

The advantage value function $A_{t+l}$ evaluates the actor-network, indicating how much the evacuation guidance agent can gain by generating response $o_{t+l}$ when in query $q$ than the average future expected benefit of the strategy. And the actor-network model iterates as follows:

$$\theta_{new} = \theta_{old} + \nabla_\theta \frac{1}{D} \sum_{i=1}^{D} \sum_{t=1}^{T} [log\pi_\theta(q, o_t)\hat{A}(q, o_t)] \tag{5}$$

Where $D$ represents the query distribution, $T$ represents the response length of the query. In actual training, multi-trajectory sampling is often used to approximate the $D$ and $T$ in a gradient of the actor-network and iterate.5 indicates that the gradient direction of the actor-network will iterate along the direction of the action with a larger advantage function.

To ensure that each strategy update of the actor-network does not deviate too much from the original strategy, the difference between the new strategy and the old strategy is not too large. In addition to setting a smaller learning rate, the PPO algorithm also introduces an importancence score and gradient truncation technique, a special objective function, which contains a truncation ratio factor to limit the ratio of the new strategy to the old strategy, converting the last part of equation 5 into:

$$\max_\theta \quad \mathbb{E}_{\tau^* \sim \pi_{\theta_{rfl}}} \min(\frac{\pi_\theta(\tau^*)}{\pi_{\theta_{rfl}}(\tau^*)}\hat{A}(\tau^*), \mathrm{clip}((\frac{\pi_\theta(\tau^*)}{\pi_{\theta_{rfl}}(\tau^*)}), 1-\epsilon, 1+\epsilon)\hat{A}(\tau^*)) \tag{6}$$

### A.3 Proof of Tool-ReRL objective

Unlike standard PPO, where the denominator of the importance weight comes from the old policy $\pi_{\theta_{old}}$, we instead use a reflection-based behavior policy $\pi_{\theta_{rfl}}$, which better captures the distribution from which the augmented trajectories are drawn. This allows Tool-ReRL to leverage reflection-derived trajectories while maintaining training stability under off-policy conditions. Equation 2 shows the resulting objective, which preserves the rich supervision signals embedded in reflections while mitigating bias introduced by the augmented distribution. Equation 7 shows how to derive the objective of Tool-ReRL by reweighting reflection-based trajectories from a reflective policy, enabling stable optimization.

$$\mathcal{J}_{ppo}(\theta) = \mathbb{E}_{\tau^* \sim \pi_{\theta_{rfl}}}[\frac{\pi_\theta(\tau^*)}{\pi_{\theta_{rfl}}(\tau^*)}\hat{A}(\tau^*)] \tag{7}$$

The detailed proof is as follows:

*Proof.*

$$\mathcal{J}_{ppo}(\theta) = \mathbb{E}_{\tau^* \sim \pi_{\theta_{old}}}[\frac{\pi_\theta(\tau^*)}{\pi_{\theta_{old}}(\tau^*)}\hat{A}(\tau^*)] = \int_{\tau^*} \frac{\pi_\theta(\tau^*)}{\pi_{\theta_{old}}(\tau^*)}\pi_{\theta_{old}}(\tau^*)\hat{A}(\tau^*)d\tau^* \tag{8}$$

$$= \int_{\tau^*} \frac{\pi_\theta(\tau^*)}{\pi_{\theta_{old}}(\tau^*)}\frac{\pi_{\theta_{old}}(\tau^*)}{\pi_{\theta_{rfl}}(\tau^*)}\pi_{\theta_{rfl}}(\tau^*)\hat{A}(\tau^*)d\tau^* = \mathbb{E}_{\tau^* \sim \pi_{\theta_{rfl}}}[\frac{\pi_\theta(\tau^*)}{\pi_{\theta_{rfl}}(\tau^*)}\hat{A}(\tau^*)]$$

## B  Benchmark and Variant Definitions

We provide concise definitions of the benchmark variants used in Table 1.

**RoTBench.** Following the official RoTBench design Ye et al. (2024), we report results on its five standard subsets: **CLE** (clean level), **SLI** (slight level), **MED** (medium level), **HEA** (heavy level), and **UNI** (union level). Each subset is evaluated under **Par.** (Parameter identification) and **Content filling** (parameter content correctness), consistent with the benchmark protocol.

**TaskBench.** We follow TaskBench Shen et al. (2024) and report the two task families: **HF** (HuggingFace tasks) and **Mm** (multimedia tool tasks). Each family includes four structural metrics: node F1 ($n\_f1$), tool/type F1 ($t\_f1$), parameter value F1 ($v\_f1$), and parameter name&value F1 ($l\_f1$).

**Seal-Tools.** Seal-Tools Wu et al. (2025b) reports **Total** (in-domain) and **OOD** performance, where OOD corresponds to the official `out of domain` subsets.

**BFCL.** We follow BFCL Patil et al. (2025), which contains **Non-Live** and **Live** variants. Accuracy is used as the official metric.

## C  REFLECTION SETUP

**Reflection Loop.** We adopt a four-stage reflection loop tailored for tool-use scenarios, enabling the model to self-correct through structured critique and controlled retry.

**Step 1: Generation.** The model first produces an initial tool call. For example, given the user query:

> "I'm considering investing and I'd like to know what's happening in the market right now. Could you get me the top market trends in the US?"

Ground-truth tool call:

```
[Market(trend_type="MARKET_INDEXES", country="us", language="en")]
```

Model output:

```
[Market(trend_type="MARKET_INDEXES", country="us")]
```

**Step 2: Critique.** A frozen critic model evaluates the generated call using the standard `Score`/`Analysis` template. For a well-formed call, the critic may return:

```
Score: Positive
Analysis: The tool call is well-formed and
uses supported parameters.
```

**Step 3: Reflection-Guided Retry.** If the critic assigns a Negative score, the system extracts the critic's analysis and converts it into a concise reflection describing missing or misused parameters. For instance, if an optional parameter is omitted:

```
Score: Negative
Analysis: The parameter 'trend_type' is valid, but the call
is missing the optional field 'language',
which should be specified for this endpoint.
```

The failure history and reflection are then embedded in the next retry prompt:

```
Your previous attempt had issues. History:
Attempt 1: [Market(trend_type="MARKET_INDEXES", country="us")]
Feedback: Missing optional parameter 'language',
which this endpoint expects.

Original request: "I'm considering investing...
top market trends in the US."
Please try again.
```

The model retries until a Positive score is obtained or a maximum retry limit is reached.

The full critic instruction is shown below:

```
You are a critic agent tasked with assessing whether the response's
reasoning process and agent response align with the ground truth.
When the response involves tool usage, ensure that its final output
constitutes a valid tool call.
Provide a consistent and objective evaluation.
```

```
Format your answer as follows:
Score: positive or negative
Analysis: Provide a single sentence explanation detailing why the
response is effective or ineffective, avoiding bullet points.
Please be concise in your reasoning.
```

**Evaluation Prompt.** To supply the critic with context for scoring, we pass the model response together with the corresponding ground truth. The critic receives the following evaluation template:

```
Analyze this interaction:
    agent response: {response}
    the ground truths of the questions: {ground_truth}
```

