# OpenReview forum: "Reflective Reinforcement Tool Learning"
_ICLR.cc/2026/Conference — ICLR 2026 Conference Withdrawn Submission_

### Official Review · Reviewer_sJZG · 2025-10-28

**Soundness:** 2
**Presentation:** 3
**Contribution:** 2
**Rating:** 2
**Confidence:** 4

**Summary:**

This paper investigates reinforcement learning for large language models in tool-use scenarios. The authors propose a new off-policy framework Tool-Reflective Reinforcement Learning with two key designs: (i) a reflection mechanism to adjust the failure trajectory during the rollout; (ii) an importance sampling estimator to bridge the gap between the current policy and the one that guides the reflection.

**Strengths:**

- This paper is well-originalized and clearly written.
- The motivation is well-established. Specifically, this paper seeks to improve the data efficiency during reinforcement learning in tool-LLM.

**Weaknesses:**

- The authors propose using samples with reflective responses, which may introduce distributional shifts. I recommend that the authors provide experimental validation, such as a distributional comparison, to make the argument more convincing.
- The description of **Assumption 1** is vague. Does it imply that $\pi_{\theta_{rfl}}(\tau^\*|q^\*) = \pi_{\theta}(\tau^\*|q^\*)$? If so, the use of importance sampling would be unnecessary.
- Based on the experimental results (e.g., **CoT** vs. **CoT + IS**), importance sampling does not appear to significantly improve performance. This point requires further explanation, even though the proposed method achieves notable improvements.
- The paper lacks a concrete training example to illustrate how the reflection mechanism is designed.
- The overall contribution of this work is limited. For example, importance sampling is a standard component of PPO, and the paper does not offer substantial changes.

**Questions:**

See them in Weaknesses.

---

> ### Author Response · Authors · 2025-11-24
>
> # Q1 Distributional comparison
> We agree with the reviewer that reflection-based query augmentation introduces a distributional shift, since the augmented prompt differs from the original query. While this shift is indeed observable in practice, we acknowledge that the current draft does not include an explicit quantitative comparison.
>
> In the revised version, **we will include an empirical analysis that measures the distributional differences between the original and reflection-augmented queries** This will make the presence and magnitude of the shift clearer and help substantiate the necessity of our IS correction step.
>
> # Q2 Formula
> $\pi_{\\theta_{\\text{rfl}}}(t_i^{*}, a_i^\* \\mid q_i^\*)$ $\\approx$
>
> $\pi_{\\theta}(t_i^{*}, a_i^\* \\mid q_i^\*)$
>
> **is not we state in the paper, in the paper we states as**
>
> $\pi_{\\theta_{\\text{rfl}}}(t_i^{*}, a_i^\* \\mid q_i)$ $\\approx$
>
> $\pi_{\\theta}(t_i^{*}, a_i^\* \\mid q_i^\*)$
> There exists a parameter configuration such that the trajectories generated by the model under the original query match (or closely approximate) those generated under the reflection-augmented query.

---

> > ### Author Response · Authors · 2025-11-24
> >
> > ### Q3 CoT vs. CoT + IS
> >
> > We agree that the performance gap between CoT and CoT+IS is small. **This is expected: IS primarily serves to stabilize training when the input prompt is modified**, and the CoT prefix introduces only a very minor change to the prompt distribution. Consequently, the distributional shift that IS is correcting is negligible, so only a small effect is observed. **The substantial improvements come from the reflection mechanism**, which injects explicit corrective information rather than merely adding a prefix; the gains are therefore attributable to reflection, not to IS alone.
> >
> > ### Q4 Reflection mechanism work through
> > Thanks for pointing out, **we will add a concrete walkthrough example would make the reflection mechanism clearer in the appendix of the revised version**, detailing the inputs and outputs. This should address the ambiguity in the current description.
> >
> > ### Q5
> >
> > While importance sampling (IS) is indeed a standard component of PPO, the key point is that our method uses IS in a **fundamentally different role.** In vanilla PPO, IS is applied only to correct the small distributional drift caused by parameter updates. In contrast, our IS correction is specifically designed to address the prompt-conditioned distribution shift introduced by reflection. Because reflection modifies the input query, the resulting trajectories are drawn from a behaviour distribution that differs from the original one something standard PPO does not and cannot correct for.
> >
> > **It is also important to clarify that IS itself is not the main source of performance improvement. The dominant gains come from the reflection mechanism,** which provides explicit corrective information that substantially improves the likelihood of generating valid trajectories. The purpose of **IS is to ensure that these reflection-augmented trajectories can be safely reused without inducing harmful off-policy bias.**  With this correction in place, reflection and PPO interact synergistically, as confirmed by our ablation results.

---

### Official Review · Reviewer_e2tm · 2025-10-30

**Soundness:** 2
**Presentation:** 3
**Contribution:** 2
**Rating:** 2
**Confidence:** 3

**Summary:**

The paper demonstrates that fixing negative samples during the RL training process with an importance sampling to offset the distribution mismatch between refined trajectory and existing trajectory can improve the model's tool learning performance.

* It shows many negative samples generated are almost positive samples with minor correction.
* It proposes a way to fix the negative sample issue by invoking an additional model call to point out the error with suggested edit and ask the model being trained to regenerate a new trajectory based on existing input, original output and the reflection.
* It shows using the reflection based behavior policy in the importance sampling function can resolve the degradation from the new generated trajectory and further improve the performance.

**Strengths:**

* The paper performs extensive experiments on various tool use dataset including RotBench, TaskBench, BFCL and Seals with different training approaches.
* The reflection-recovery strategy is clearly explained and the overall approach is easy to understand.

**Weaknesses:**

* The detailed setup for the reflection module used in the experiment is missing.
* The ablation should include training with positive samples only as well to show the increment is not from removing negative samples and training with same number of positive samples after refinement to show the  increment is not from more training signals.
* Experiment on PPO + IS should also be included.
* Details of the reflection module are missing, such as the actual module used, what’s the input/output.
* The results from 2 model on the distribution shift does not provide strong evidence that additional refinement will cause a performance drop given one of the results does not show improvement.
* The table 1’s benchmark variant is not defined.
* Using the same policy to generate a refined trajectory should still consider on-policy. The distribution shift is not due to mixing off-policy data with on-policy data but rather mixing different task’s data
* The analysis on data with near-success or environment-related issues in a single dataset is not extensive to support the claim that many failures can be recovered through simple retries, reasoning, or reflection.

**Questions:**

* What’s the efficiency cost with additional refinement module call.
* Do all the CoT and Ref variants in all experiments only apply to negative samples?

---

> ### Author Response · Authors · 2025-11-24
>
> ### Wk3. Experiments on PPO + IS
>
> **We clarify that importance sampling is required only when the input query is modified; therefore, in standard PPO without reflection, “PPO + IS” is effectively identical to vanilla PPO.** We appreciate the reviewer’s suggestion and are happy to elaborate.
>
> In our framework, IS correction is applied solely to address the distributional mismatch introduced by query augmentation. When the policy always conditions on the original query as is the case in standard PPO there is no shift in the input distribution and thus no behavior–target mismatch that requires correction. Consequently, “PPO + IS” reduces exactly to vanilla PPO in this setting.
>
> Importantly, the IS mechanism referenced here is not the intrinsic ratio used in PPO’s policy update. Instead, it refers to an **additional IS correction** introduced in our method exclusively to compensate for query-level modifications. We will make this distinction clearer in the updated version.
>
> ### Additional Clarifications
>
> We appreciate the reviewer for highlighting these missing details, and we agree that both the reflection module and the benchmark definitions should be described more explicitly.
>
> In the revised version, **we will add a dedicated subsection specifying the reflection module’s configuration, including its input format, output format, the exact model used, and the full setup details. We will also adding the descriptions of the benchmark variants in Table 1 and adopt more precise and consistent terminology to avoid ambiguity.**

---

> > ### Author Response · Authors · 2025-11-24
> >
> > ### Q1  Cost
> >
> > **Our overall position is that the additional refinement step introduces negligible overhead relative to the savings it enables in terms of effective rollout usage.** The refinement module is invoked only after a failed attempt and produces a very short textual hint typically a few tokens—rather than a second full-length trajectory. Importantly, what dominates the computational cost in PPO-style training is not the hint itself but the number of long rollouts required to obtain a successful sample.
> >
> > In standard PPO, a trajectory contributes a meaningful gradient update only if it is correct. When the first attempt fails, the entire rollout yields no usable training signal, and PPO must resample until it eventually produces a valid outcome. For tasks with sparse rewards and low Pass@1, this can require several full rollouts per query before any positive advantage is observed.
> >
> > By contrast, our refinement mechanism substantially increases the likelihood that the next rollout succeeds. After a failure, the model receives a brief diagnostic hint that helps correct the error in the next attempt, often enabling a valid trajectory in one retry. As a result, the expected number of full-length rollouts needed to obtain a successful sample is significantly reduced. Even though we add a short refinement call, the reduction in repeated long rollouts more than compensates for this overhead.
> >
> > ### Q2  Apply to negative samples?
> > **Yes—the CoT and Reflection variants are applied only to failed trajectories, by design.**
> >  We thank the reviewer for raising this point. Positive trajectories already yield valid tool calls and therefore do not require additional reasoning or hints to succeed. Applying CoT or Reflection to such trajectories would introduce unnecessary distributional shifts without providing meaningful benefit.
> >
> > Instead, CoT or Reflection is invoked only when a failure occurs: Reflection provides failure-specific contextual information to guide a corrected retry, while the +CoT variant tests whether adding self-generated reasoning can help recover an otherwise unsuccessful attempt. Successful samples remain unchanged to preserve the original distribution and avoid altering trajectories that already provide correct policy signals.

---

### Official Review · Reviewer_ysiG · 2025-11-01

**Soundness:** 3
**Presentation:** 4
**Contribution:** 2
**Rating:** 2
**Confidence:** 4

**Summary:**

This paper introduces Tool-Reflective Reinforcement Learning (Tool-ReRL), a framework that addresses data inefficiency in reinforcement learning for tool use by large language models. The authors identify that 44.7% of failed trajectories in standard RL training are near-success cases discarded due to environmental perturbations rather than policy errors. Tool-ReRL incorporates a reflection mechanism that analyzes failures and generates corrective feedback, combined with importance sampling to handle the distributional shift from reflection-augmented trajectories. Experiments on four benchmarks using Qwen-2.5-7B and LLaMA-3.1-8B models demonstrate performance gains of up to 7.60% and 6.11% respectively over standard RL algorithms, with the framework consistently outperforming supervised fine-tuning, DPO, and PPO baselines. The results show that both reflection and importance weighting components are necessary - reflection alone decreased performance by 3.4% without proper correction, while the complete framework transforms previously wasted near-success trajectories into valuable training signals, improving data efficiency in tool learning scenarios where environmental instabilities are common.

**Strengths:**

1. Good problem statement, data efficiency in RL training is an important problem
2. Fixing issues with tools or minor changes to trajectory as correction is a good idea and makes sense
3. Comparison against a good set of baselines and a broad set of tool use benchmarks
4. Importance weighting is clearly motivated, explained, and evaluated with ablations
5. Presentation is clear and easy to follow

**Weaknesses:**

- One of the baseline methods should be on-policy distillation [1]. The "reflection" model is much more powerful than the model being trained. Such a "teacher" model is not available in SFT or PPO. And use of distillation will likely be more efficient than PPO.
- The paper claims that the method is more data efficient. But there is no measurement of how efficient the proposed method is. How much less number of epochs does the proposed method need to reach the same score as the baseline models?
- The paper lacks a measure of computational efficiency. A much larger model (Deepseek R1) is used as a reflection model. It is possible that the compute and time used to create such reflections exceeds the time it takes to recover from discarding erroneous data.




[1] Agarwal, Rishabh, et al. "On-policy distillation of language models: Learning from self-generated mistakes." The twelfth international conference on learning representations. 2024.

**Questions:**

1. "we identified 834 failures in total, of which 373 (44.7%) were attributable to near-success or environment-related issues." How do you identify "near-success"?
2. How do you verify that the reflection model is correct? What happens when it is incorrect?
3. Why is this method only limited to "near-success" trajectories? Why not apply it to all failed trajectories?
4. "if the policy lacks sufficient capability to sample positive examples with reasonable success rates, its contribution
to RL effectiveness becomes negligible." What is the evidence for this claim?
5. "tool invocation does not require long-form reasoning" -- what is the evidence for this claim?
6. I did not understand this: "+CoT+IS allows the model to generate its own internal thought sequence but does not perform reflection, while applying importance weights to reduce distributional discrepancy."
7. Paper claims PPO halves the compute required compared to DPO. PPO requires use of a critic model, which is very expensive. Can you substantiate this claim?

---

> ### Author Response · Authors · 2025-11-24
>
> ### W1. Use of a stronger external model for reflection
>
> Our main clarification is that the core Tool-ReRL experiments **do not rely on a stronger external model for generating reflections. The policy model produces its own reflections.**  We sincerely thank the reviewer for the constructive suggestion and the opportunity to clarify this point.
>
> In our primary setting, reflection is generated directly by the policy itself, so the framework does not operate in a teacher–student or distillation paradigm. The stronger model **(DeepSeek-R1) appears only in Table 2 as an ablation**. Its sole purpose is to show that reflections originating from the policy yield larger performance gains than those produced by a stronger external model. This comparison highlights the robustness and self-sufficiency of our approach rather than any dependence on a more capable teacher model.
>
> ### W2 & W3. Data efficiency and computational cost
>
> Our central clarification is that “data efficiency” in our work refers to **achieving higher performance under the same number of RL updates** and the **same amount of collected interaction data**. The data budget is held fixed; what differs is how effectively that budget is used.
>
> Standard PPO discards failed trajectories because they provide no positive learning signal. In contrast, our method recovers a substantial portion of these failures by converting them into usable trajectories through reflection. Consequently, a larger fraction of collected data contributes meaningful gradients without requiring any additional environment interaction. Under the same number of epochs or environment steps, this higher yield of useful training signals leads to faster improvement compared with PPO baselines—this is the intended notion of data efficiency in our experiments.
>
> **Our main argument regarding computational cost is that *in expectation* the number of full-length rollouts required to obtain a positive training signal is lower under our reflection mechanism than under standard PPO, even after accounting for the additional hint step.** For any query, if the initial rollout fails, the entire trajectory is unusable in PPO. Thus, to obtain a single correct trajectory, PPO often must regenerate multiple full-length rollouts, incurring substantial token usage and wall-clock time.
>
> In contrast, our method obtains correctness with **significantly fewer expected long rollouts**. Upon failure, we generate a short reflection hint and issue one guided retry. The reflection substantially increases the success probability of the next attempt, reducing the expected number of full-length trajectories required to obtain a correct sample.

---

> > ### Author Response · Authors · 2025-11-24
> >
> > ### Q1 and Q3  How do you identify “near-success
> >
> > **We define “near-success” as failure cases where the model’s high-level reasoning and tool selection are already correct, and the final tool call fails only due to small, locally correctable errors.** Typical examples include slightly mis-specified fields (e.g., an incorrect type or a missing optional argument) or minor formatting inconsistencies. In these cases, the tool choice and reasoning chain are sound—the trajectory fails only at the final detail.
> >
> > To identify such cases, we replay all 834 failed trajectories with their corresponding reflection hints and check whether the reflection-guided retry yields a valid tool call. Among them, 373 trajectories (44.7%) are successfully repaired, indicating that their failure mode stems from small, fixable issues rather than deeper reasoning mistakes.
> >
> > **We apply reflection only to these near-success cases because they can be reliably turned into valid positive examples.** In contrast, failures involving incorrect tool choice or fundamentally wrong parameter selection cannot be corrected through reflection alone; applying reflection to such cases does not improve success rates and may introduce noise. Restricting reflection to recoverable failures ensures that the method remains both efficient and effective.
> >
> > ### Q2 How do you verify that the reflection model is correct?
> >
> > **We emphasize that our method does not require the reflection model to be correct, because reflection is never used as a supervision target** it only serves as a lightweight prompt modifier to encourage resampling. The policy’s actual action is always generated by the trained model itself, and the environment reward is the only criterion for deciding whether a reflection-augmented trajectory is incorporated into learning.
> >
> > If a reflection happens to be incorrect or misleading, the resulting tool call simply fails again and receives zero reward, making the trajectory unusable and contributing nothing to the update. Thus, the correctness of the reflection text is neither assumed nor required; the method remains reliable because only trajectories validated by the environment signal enter the learning process.
> >
> > ### Q5 evidence for tool invocation does not require long-form reasoning
> > In the specific benchmarks we evaluate, **successful tool invocation primarily depends on producing the correct tool call rather than generating long-form reasoning traces.** We thank the reviewer for raising this question.
> >
> > In the benchmarks used in our experiments, RoTBench, BFCL (simple), and TaskBench the tasks typically require only short reasoning to identify the appropriate tool and its arguments. As a result, long-form reasoning is not necessary for achieving high performance on these datasets.
> >
> > ### Q7 DPO and PPO
> >
> > We appreciate the reviewer for catching this, and we clarify that **our intended point concerns data requirements rather than wall-clock compute.**  Our original phrasing was imprecise, we did not mean to suggest that PPO universally halves computational cost relative to DPO.
> >
> > The distinction we intended to highlight is that, for the same set of queries, DPO typically requires roughly twice as many labeled samples because each query must produce both a preferred and a rejected trajectory, whereas PPO operates on single trajectories with scalar rewards. In contrast, PPO does not require paired data and therefore has a lower data requirement per query. We fully agree that PPO introduces its own computational overhead through the critic, and thus the comparison should not be framed in terms of raw wall-clock time. In the camera-ready version, we will revise the statement to the more accurate and data-centric formulation:
> > **DPO requires roughly twice as many samples per query as PPO, since it needs both successful and failed trajectories.**

---

> > > ### Author Response · Authors · 2025-11-24
> > >
> > > ### Q4  PPO Claim
> > >
> > > **This claim follows directly from how PPO updates are computed in sparse-reward settings**. PPO’s policy-gradient update is driven by the advantage term $(A_t = r_t + \gamma V(s_{t+1}) - V(s_t))$, which becomes positive only when the policy actually samples a successful trajectory.
> > >
> > > If the policy rarely produces valid tool calls, then (i) all returns are zero, so the value function receives no positive signal to bootstrap, and (ii) all sampled trajectories have non-positive advantage. In this regime, PPO performs almost no meaningful update: penalizing failures (negative or zero advantage) does not move the policy toward the unknown correct action manifold, because there is no gradient signal indicating *which* actions would lead to success.
> > >
> > > This phenomenon is well-documented in sparse-reward RL and has been repeatedly observed in prior tool-use work, where initial Pass@1 rates are extremely low. Thus, without a non-negligible probability of sampling successful trajectories, PPO cannot effectively improve the policy because it lacks positive-advantage examples to learn from.
> > >
> > > ### Q6 definition of COT+IS
> > >
> > > **The key distinction is that reflection augments the query using information extracted from a previous failed attempt, whereas +CoT+IS augments it only with a self-generated reasoning trace that has no access to failure context.**  We sincerely thank the reviewer for pointing out the ambiguity, and we are happy to clarify this distinction.
> > >
> > > In the reflection variant, the model receives a brief hint that explicitly states what went wrong in the previous rollout. Thus, the augmented query contains failure-specific diagnostic information that directly guides the model toward repairing the trajectory.
> > >
> > > In contrast, the +CoT+IS baseline augments the query solely with a short, self-generated chain-of-thought derived from the original query. It does not incorporate any information about prior failures. Because this CoT-augmented query differs slightly from the original one, it introduces a small distributional shift; accordingly, we apply importance-sampling weights to correct for the mismatch between the modified query and the original behavior policy.

---

> > > > ### Comment · Reviewer_ysiG · 2025-11-25
> > > > **Sparse reward**
> > > >
> > > > Thank you, both these responses makes sense.
> > > >
> > > > A small correction:
> > > > "penalizing failures (negative or zero advantage) does not move the policy"
> > > > Negative advantages absolutely help with improving policies. There is a change in loss value, which directly impacts the policy. For zero advantage, loss is zero, which will not change the policy.

---

> > > ### Comment · Reviewer_ysiG · 2025-11-25
> > > **Near success, reasoning efficiency**
> > >
> > > Thank you for the clarification on the near success. I would have framed it as only including trajectories that were corrected with reflection. It is an observation that such trajectories were "near-successes", but it is not a mathematical formulation or an algorithm. In any case, the design of the algorithm makes sense now.
> > >
> > > You need to quantify that tool call does not require long-form reasoning by measuring the mean number of reasoning tokens required for tool calls. The qualifying statement "the tasks typically require only short reasoning" is vague and unsupported.

---

> > ### Comment · Reviewer_ysiG · 2025-11-25
> > **Reflection model and computational cost**
> >
> > Thank you for clarifying that the reflection model is the policy itself. That makes this a viable algorithm, and it cannot be compared with policy distillation.
> >
> > I understand the point you are making with data and computational efficiency. In my opinion, it is not sufficient to make the data efficiency argument. You need to measure the computational cost of the proposed the algorithm and include it in the paper. Even for data efficiency, you need to quantify the percentage savings compared to baseline algorithms rather than make a qualifying statement such as "significantly fewer expected long rollouts".

---

> ### Comment · Reviewer_ysiG · 2025-11-25
> **Overall score**
>
> I've changed my score from 2 to 4 based on the responses.
>
> To me, it is critical that the data efficiency and computational cost of the algorithm is quantified before the paper can be accepted.

---

### Official Review · Reviewer_T1Ru · 2025-11-04

**Soundness:** 3
**Presentation:** 3
**Contribution:** 2
**Rating:** 4
**Confidence:** 2

**Summary:**

The paper proposes Tool-ReRL, an off-policy RL framework for tool learning that repairs failed rollouts online via a reflection module and re-uses the repaired trajectories with importance-weighted correction inside a PPO objective. This design converts many environment-induced or near-success failures—typically discarded by prior work—into useful training signals while controlling distributional drift. On four tool-use benchmarks, Tool-ReRL consistently improves average performance over strong baselines with the same data budget (up to +7.60% / +6.11% on Qwen2.5-7B / LLaMA3.1-8B).

**Strengths:**

- The motivation is clear and reasonable. Environment perturbations create many “false negatives”; turning them into learnable signals can improve the sampling efficiency.

- The analysations and ablations are strong and comprehensive.

**Weaknesses:**

- The method hinges on the inverse-prompt equivalence to estimate behavior policy for IS correction. However, quantitative (or theoretical) bias analysis of the assumption is omitted.

- Computational cost & scalability are not carefully analysed. Reflection attempts and extra sampling add non-trivial wall-clock costs. The paper should report env-steps / wall-clock time / money against strong RL baselines under matched budgets.

**Questions:**

See weaknesses.

---

> ### Author Response · Authors · 2025-11-24
>
> ### W1. Behavior-policy approximation assumption
>
> **Our central point is that the policy shift induced by using q\* is minimal, and any resulting bias remains strictly bounded by the clipped IS ratio.** We sincerely thank the reviewer for the thoughtful and perceptive comments regarding both our assumption and the potential overhead introduced by reflection. These concerns offer a valuable opportunity to clarify our design choices.
>
> Regarding W1 specifically, we note that q\* differs from q only through a short textual suffix, which in practice induces only a small change in the conditional distribution. Empirically, we consistently observe low KL divergence between π(·∣q) and π(·∣q\*). Any remaining mismatch is further constrained by the clipped IS ratio, ensuring that the effect of an imperfect behavior-policy approximation stays strictly limited.
>
> ### W2 Computational cost and scalability
>
> **Our main point is that, in expectation, obtaining a positive training signal under our reflection mechanism requires fewer sampled rollouts than standard PPO, even accounting for the additional hint step**. In standard PPO, a query contributes a useful gradient update only when the model happens to produce a correct trajectory; an incorrect rollout yields no learning signal. Consequently, PPO often requires repeatedly generating full-length trajectories until a correct one appears a process that is costly in both token usage and wall-clock time.
>
> In contrast, our method reaches correctness with substantially fewer expected long rollouts. When a failure occurs, we generate a short reflection hint (only a few tokens) and issue a single guided retry. The reflection meaningfully increases the success probability of the second attempt, substantially lowering the expected number of full-length trajectories required to obtain a correct sample. As a result, the overall computational cost is reduced despite the extra hint step.

---

### Note · Authors · 2026-01-05

I have read and agree with the venue's withdrawal policy on behalf of myself and my co-authors.